# Impact of a Breathing Intervention on Engagement of Abdominal, Thoracic, and Subclavian Musculature during Exercise, a Randomized Trial

**DOI:** 10.3390/jcm10163514

**Published:** 2021-08-10

**Authors:** Petr Bahenský, Václav Bunc, Renata Malátová, David Marko, Gregory J. Grosicki, Jan Schuster

**Affiliations:** 1Department of Sports Studies, Faculty of Education, University of South Bohemia, 371 15 České Budějovice, Czech Republic; malatova@pf.jcu.cz (R.M.); David.Marko@seznam.cz (D.M.); jan.schuster@seznam.cz (J.S.); 2Sports Motor Skills Laboratory, Faculty of Sports, Physical Training and Education, Charles University, 165 52 Prague, Czech Republic; Bunc@ftvs.cuni.cz; 3Department of Health Sciences and Kinesiology, Biodynamics and Human Performance Center, Armstrong Campus, Georgia Southern University, Savannah, GA 31419, USA; ggrosicki@georgiasouthern.edu

**Keywords:** breathing pattern, breathing exercise, load, diaphragm, adolescents

## Abstract

Background: Breathing technique may influence endurance exercise performance by reducing overall breathing work and delaying respiratory muscle fatigue. We investigated whether a two-month yoga-based breathing intervention could affect breathing characteristics during exercise. Methods: Forty-six endurance runners (age = 16.6 ± 1.2 years) were randomized to either a breathing intervention or control group. The contribution of abdominal, thoracic, and subclavian musculature to respiration and ventilation parameters during three different intensities on a cycle ergometer was assessed pre- and post-intervention. Results: Post-intervention, abdominal, thoracic, and subclavian ventilatory contributions were altered at 2 W·kg^−1^ (27:23:50 to 31:28:41), 3 W·kg^−1^ (26:22:52 to 28:31:41), and 4 W·kg^−1^ (24:24:52 to 27:30:43), whereas minimal changes were observed in the control group. More specifically, a significant (*p* < 0.05) increase in abdominal contribution was observed at rest and during low intensity work (i.e., 2 and 3 W·kg^−1^), and a decrease in respiratory rate and increase of tidal volume were observed in the experimental group. Conclusions: These data highlight an increased reliance on more efficient abdominal and thoracic musculature, and less recruitment of subclavian musculature, in young endurance athletes during exercise following a two-month yoga-based breathing intervention. More efficient ventilatory muscular recruitment may benefit endurance performance by reducing energy demand and thus optimize energy requirements for mechanical work.

## 1. Introduction

The physiological implications of reductions in physical activity due to an environment that is oversaturated with technological innovation are only beginning to be realized [1,2]. Adverse changes in respiratory patterns are just one of these deleterious adaptations, and dysfunctional breathing has become increasingly common with an expected prevalence of between 60–80% in otherwise healthy adults [3].

Gas exchange during normal activity is coordinated by inspiratory and expiratory processes involving synchronized movement of the upper and lower chest, abdomen, and diaphragm [4,5,6,7]. In the resting state, breathing is regulated by an expansion of the lower chest and anteroposterior movement of the sternal bones that is facilitated by the diaphragm and intercostal muscles that account for ~2–5% of whole-body oxygen consumption at rest. During intensive muscle work, respiratory energy demand can increase several times. In the case of trained athletes, it reaches up to 10% of the total energy consumption during moderately demanding activity [8,9,10]. Meanwhile, excess involvement of areas of the upper chest distinctly characterizes respiratory inefficiency and potential breathing disorders [4,11], which may become increasingly relevant during high intensity work where ventilatory oxygen demand may comprise 15% (or more) of whole- body oxygen consumption [8,9,10].

Though inter-individual variability characterized by differential involvement of abdominal, thoracic, and subclavian body sectors in breathing patterns have been observed [12], systematic analysis of the effect of breathing technique on athletic performance is vastly understudied [13]. However, the influence of breathing patterns on performance has recently come to the forefront of physical activity research [14], and the role of the diaphragm during high-intensity work has received significant attention [13,15]. In individuals with dysfunctional respiration, the pain threshold is lowered, and control of motor functions and movement dysfunctions are impaired, all of which may adversely affect the individual′s physical performance [16]. Pertinently, increasing diaphragmatic respiratory involvement reduces breathing effort, improves ventilation efficiency, reduces dyspnea, improves exercise tolerance, and can be trained [17,18,19,20]. Thus, there is great incentive to elucidate techniques to improve respiratory efficiency as a potential means to improve athletic performance [21,22,23].

Specific respiratory (inspiratory) muscle training (IMT) improves the function of the inspiratory muscles. According to literature and clinical experience, there are three established methods: (1) resistive load, (2) threshold load, and (3) normocapnic hyperpnea. Each training method and the associated devices have specific characteristics [24]. Setting up an IMT should start with specific diagnostics of respiratory muscle function and be followed by detailed individual introduction to training. Changing respiratory muscular activity through strengthening of inspiratory muscles may attenuate disease risk. Weakness or fatigue of the diaphragm and the accessory muscles of inspiration is widely recognized as a cause of failure to wean from mechanical ventilation [25]. The influence of IMT on exercise performance has also been surveyed. Faghy and Brown [22] provided evidence for the ability of IMT training to improve exercise performance (time trial) with thoracic load carriage.

Many methods can be used to evaluate the respiratory pattern [26], the most common of which are palpation, chest circumference, plethysmography of the whole body, chest skiagram, spirometry or various instruments recording changes in height of individual torso segments, or through a three-dimensional system [27,28,29]. Estimation of chest wall motion by surface measurements allows one-dimensional measurements of the chest wall by assessment with an optical reflectance system [30] or by three-dimensional tracking [31,32]. Chest wall volume changes can be assessed by optoelectronic plethysmography [33] or by optoelectronic plethysmography [34]. Building upon these techniques, our group used a respiratory muscle dynamometer to measure instantaneous values of involvement of the ventilatory musculature (MD03 muscle dynamometer) [11,35,36]. Using this dynamometer, the present study evaluated whether two-months of a yoga breathing exercise program may influence breathing characteristics during various intensities of exercise in young healthy athletes.

Based on previous work by our group [37], we hypothesized that the breathing exercise program would modulate respiratory musculature contribution. We anticipated greater involvement of the musculature of the lower torso (i.e., abdomen and thoracic sectors) and less upper-body contribution (i.e., subclavian) during exercise following the yoga-based breathing intervention.

## 2. Materials and Methods

### 2.1. Subjects

Forty-six adolescent distance runners (14–18 years) participated in our study: 23 males (age = 16.4 ± 1.1, height = 177.1 ± 5.8 cm, weight = 62.4 ± 5.8 kg) and 23 females (age = 16.8 ± 1.1, height = 168.5 ± 4.4 cm, weight = 55.9 ± 4.0 kg). All participants reported a history of endurance running of at least six times a week for the past year. They are all members of the same training group, and thus training volume and intensity were comparable throughout the duration of the study. Participants were randomly allocated to an experimental group (*n* = 23), which took part in an eight-week breathing intervention, or a control group (*n* = 23), which continued training but did not carry out the yoga-based breathing intervention. One participant did not complete the intervention for medical reasons unrelated to the intervention and was excluded from the study. The two groups, both experimental and control, followed the same training program, the only difference being that the experimental group performed the yoga-based breathing intervention. A randomization sequence has been generated using Randomization.org. An independent person not involved in this study made the computer-generated randomization sequence. The study protocol was reviewed and approved by the local ethics committee on 19 October 2018 (002/2018) and followed the guidelines of the World Medical Assembly Declaration of Helsinki. This research is a clinical trial (NCT04950387). Written informed consent to participate was provided by guardians and verbal assessment was provided by the participants.

### 2.2. Study Design

#### 2.2.1. Measurement of Sectors Engagement

The testing took place in the Laboratory of Load Diagnostics at Department of Sports Studies, Faculty of Education, University of South Bohemia. We evaluated ventilatory musculature involvement in three basic areas (Figure 1; abdominal = red sensor, thoracic = green sensor, and subclavian = blue sensor) using a muscle dynamometer MD03 as previously described [35,36].

The device is a four-channel digital muscle dynamometer that, by design, allows instantaneous values of muscle force to be measured in relation to time (i.e., both the force size and its dynamics can be evaluated). In general, different muscles and muscle groups on the human body can be measured. MD03 is made up of four muscle probes (we used three probes) that attach themselves to the human body with belts. Greater muscle involvement in the segment of interest at a higher tidal volume results in higher dynamometer pressure values. The probes contain a strain transducer to a digital signal that is transmitted to a microprocessor evaluation unit that adjusts digital signals from the probes into a compatible form with a USB input to a notebook. Probe attachment sites were selected based on the kinematics of the aforementioned thoracic sectors.

The first probe was placed in the lower respiratory sector on the ventral side of the level L4–5. The second probe was placed on the ventral side just below the sternum (between ribs 8 and 9). The third probe was characterized by upper respiratory musculature involvement and was placed between ribs 3 and 4 on the ventral side on the sternum. Chest compression and expansion during respiration change the force applied to the individual sensors in the attached belt.

#### 2.2.2. Measurement of Ventilatory Parameters

Inspiratory and expiratory forces exerted on individual probes located in the given breathing sectors were recorded for 60 s and minute averages were determined for each probe. After 60 s of resting data acquisition using both spontaneous and deep breathing, participants underwent an incremental test on a cycle ergometer (Lode, Groningen, The Netherlands) and oxygen consumption, tidal volume, respiratory rate, and minute ventilation were continuously monitored (Metalyzer B3, Cortex, Leipzig, Germany). The exercise protocol consisted of a graded exercise test that was made relative to participant body weight (i.e., W·kg^−1^) and began with a 4-minute stage at 1 W·kg^−1^ followed by three, two-minute stages (for partial stabilization of ventilation parameters) at progressive intensities (2, 3, 4 W·kg^−1^) as we have described previously [38,39] and cadence was standardized to 95–100 rev·min^−1^. Ventilatory muscular involvement of the abdominal, thoracic, and subclavian body sectors was monitored during the last minute of each of the three submaximal intensities.

#### 2.2.3. Breathing Exercise Program

The training program lasted eight weeks. The experimental group performed yoga-based breathing exercises daily. In the first week of the breathing intervention, training took place in the form of three supervised group sessions [37]. In the following weeks, there were always two group training sessions, each lasted ~30 min. On unsupervised days, participants were asked to perform exercises individually at home for at least 10 min. Information about the length of each individual’s training session was recorded in a diary by the participant.

The design of the breathing exercise program was based on yoga, and the aim was to activate the diaphragm and become aware of individual breathing sectors. As such, breath training included a variety of exercises, such as breathing wave training, full breathing (breathing into all sectors), and paced breathing (breathing in a specified rhythm). The exercises were performed in various positions, including lying down, sitting in the kneeling position, sitting, kneeling, and standing (see Appendix A). All breathing was performed through the nose. At the beginning of the intervention, the participants breathed spontaneously, later switching to prolonging the inspiratory and expiratory phases. They started with a 1:1 ratio of inhale to exhale length. Gradually, the pre-exhalation and pre-exhalation phases of breath holding were included: inspiration-6 periods, holding breath-3 periods, exhaling-6 periods, holding breath-3 periods. Each of the participants adapted the exercise to their individual respiratory rate. Each of the exercises was repeated six times. The exercises were slow, with a deep focus on breathing, in line with the movement. Very important was the perception of the direction of movement and expansion of the chest, the behavior of the axis of the body (head, spine, pelvis), which they learned during the introductory meetings [37]. The control group did not participate in any form of breathing training and were told to go about their lives as usual.

The follow-up testing, which was the same as the aforementioned described graded maximal test on the cycle ergometer, was performed after eight weeks of intervention.

### 2.3. Statistical Analysis

Data are presented as mean ± SD. The normality of data was confirmed using the Shapiro–Wilk test. A two-way repeated-measures ANOVA (group × intensity) was used to compare changes in the involvement of individual breathing sectors and respiratory rate in the intervention and control groups. Significant interactions were examined using Bonferroni adjusted simple main effect post hoc comparisons. An alpha-level of 0.05 was used to assess statistical significance for all comparisons. Subsequently, effect size was determined using Cohen′s d. The Pearson correlation coefficient was used to examine relationships between changes in tidal volume and pressure values on the dynamometer. The alpha-level was set to 0.05. The data processing was done in Excel 2016 (Microsoft, Oregon, WA, USA) and Statistica 12 (StatSoft, Tulsa, OK, USA).

## 3. Results

Participants carried out the yoga-based breathing program for an average of 13.3 ± 2.8 min per day during the two-month period. In the experimental group, there was a significant increase in the involvement of the abdominal segment during deep breathing and at 2 and 3 W·kg^−1^ (*p* ˂ 0.05; see Figure 2 and Table 1). The only significant change in thoracic involvement was seen at 3 W·kg^−1^ (*p* ˂ 0.01). In subclavian respiration, there was no significant change in involvement at any of the intensities, even at rest or at rest during deep resting breathing. In the control group, there was no significant change in the involvement of individual breathing sectors at rest or at any load level (*p* > 0.05; Table 1).

As a result of the breathing exercise intervention, the experimental group experienced a significant reduction (*p* < 0.05) of respiratory rate under load 3 and 4 W·kg^−1^, with medium (4 W·kg^−1^) or small (rest, deep rest, 2 and 3 W·kg^−^^1^) effect sizes. We noted a significant increase of tidal volume at 2 W·kg^−1^, there are changes with small effect size, during all intensities of load. Minute ventilation and oxygen consumption were not significantly altered (see Table 2). The overall effect of breathing exercise intervention in all phases on changes of respiratory rate and tidal volume was confirmed at level *p* < 0.01. 

Changes in tidal volume were significantly related to abdominal probe pressure at all intensities (see Table 3).

In all intensities, greater abdominal and less subclavian percentage contribution was noted (see Figure 2).

## 4. Discussion

The primary finding of the present study was an alternation in breathing patterns at rest, and during cycling exercise at various intensities, in young healthy individuals following an eight-week breathing intervention. This finding corroborates previous research by our group in showing greater and more efficient abdominal contribution to respiration following a breathing intervention [37]. Moreover, it is worthy to note that the respiratory musculature involvement following the intervention was close to what may be recommended [4].

Physical exertion often increases the perception of respiratory effort in healthy people and leads to a feeling of dyspnea. Sports activities, be it intensive, short duration (≥85% of the maximum oxygen uptake) or less intense, longer-lasting duration (“ultramarathon” etc.) can lead to fatigue of the inspiratory and/or expiratory muscles [24]. Moreover, tired respiratory muscles impair athletic performance. During physical activity and sport, work of the respiratory muscles is compounded by greater demand for postural stabilization and movement efficiency [40]. Body stability is impaired when the respiratory muscles are tired, which can increase the risk of tripping or falling [24].

Respiratory therapy is an integral part of treatment for many patients with various diseases. Respiratory contributions have been shown to limit exercise in patients with heart failure. The manner in which the respiratory system limits exercise is due to abnormalities in ventilation, perfusion, or both ventilation and perfusion inspiratory muscle weakness may induce several impairments in both healthy and athletic individuals [22]. Similarly, studies have demonstrated that inspiratory muscle strength also has an important role in the pathophysiology of exercise limitation in several clinical conditions. Indeed, IMT is becoming an effective complementary treatment with positive effects on muscle strength and exercise capacity. More recently, studies have found that maximal inspiratory pressure (MIP) is strongly correlated with VO_2_ peak in patients after acute myocardial infarction and heart failure, reinforcing the influence of the inspiratory muscles on functional capacity [41]. The exercises primarily reduced end-expiratory lung volume rather than end-inspiratory lung volume, which is constrained by the presence of a thoracic load. Consequently, the training stimuli may be targeting and strengthening the inspiratory muscles throughout an operational range, which may not be utilised during exercise with load carriage. Importantly, previous work has identified that fatigue of the expiratory muscles is not an influencing factor in determining operational lung volumes, despite reduced end-expiratory time and increased peak gastric and esophageal pressures, and it may be more appropriate to assess influences that inhibit flow [42]. In general, the IMT performed at an intensity of 30% MIP resulted in decreased cardiac sympathetic modulation (LF) and increased parasympathetic (HF) at rest in patients with hypertension, heart failure, and diabetes mellitus [43]. Nevertheless, this measure has been questioned as interventions can elicit either complex non-linear reciprocal or parallel changes in either division of ANS, and these complex interactions can influence the calculation and interpretation of LF/HF [44]. However, applying IMT to different diseases, associated with a variety of training protocols, as well as few studies found in the literature, makes the effects of IMT on cardiovascular autonomic control inconclusive. Inspiratory muscle training promotes changes in cardiovascular autonomic responses in humans [43]. Though inspiratory muscle training seems to improve maximal inspiratory pressure, it remains unclear whether these benefits translate to weaning success and a shorter duration of mechanical ventilation [25].

It is important to note that all participants were encouraged to breathe spontaneously during the testing period to ensure that any observed changes were in fact attributable to the intervention. Interestingly, the observed significant increase in abdominal contribution to breathing was noted at rest and during light/moderate intensities (2 and 3 W·kg^−1^), but not at the greatest load (i.e., 4 W·kg^−1^). Greater involvement of the thoracic musculature was also observed at lower (i.e., 3 W·kg^−1^) but not the greatest workload. This may be ascribed to greater anaerobic energy contribution at the greatest workload and thus the need for excess ventilation to remove rapidly accumulating CO2 [13,45]. However, of relevance was the observation of a trend towards a reduction in subclavian involvement at the higher workload in the experimental group. At the same time, there was a decrease in respiratory rate, an increase in tidal volume while maintaining the minute ventilation volume and oxygen consumption. Reduced subclavian involvement together with decreased respiratory rate and increased tidal volume at the same minute ventilation volume and same VO_2_ denotes greater respiratory efficiency and thus greater oxygen availability for mechanical musculature. At the same time, a decrease in respiratory rate also signals a decrease in respiratory work as one of the possible effects of a targeted breathing exercise program. A link has been shown between an increase of pressure on the abdominal probes and an increase of tidal volume. Furthermore, the increased contribution of the abdominal sector to respiration, together with the decreased respiratory rate, and increased tidal volume with the similar minute ventilation, indicates an improved breathing economy [38,46]. This is important as respiratory muscle efficiency is one of the conditions for good performance in endurance.

At rest, and during deep breathing, greater recruitment of abdominal muscles helps to optimize respiratory efficiency and delay the onset of respiratory muscle fatigue. However, during submaximal exercise, a significant alteration in respiratory musculature characterized by a reduction in abdominal and increased subclavian contribution is observed. Our results suggest that while it is possible to manipulate spontaneous breathing patterns during exercise, these benefits may be limited to lighter loads that are likely below the ventilatory threshold. However, reduced respiratory rate following breath training at both low and high workloads may be of benefit across a range of exercise performance disciplines.

The present findings should be interpreted in the context of the population; a greater training effect may be anticipated in adolescents in whom respiratory patterns during exercise are not as well engrained [47]. Like previous research in the field, we strategically selected an eight-week training intervention [48,49,50,51]. Future studies to determine the possible benefits of shorter breath training interventions, as well as the persistence of these adaptations if breath training is stopped, are warranted. Other limitations of the present work include our relatively modest sample size as well as that much of the training was performed at-home without direct supervision. Verification of these findings in non-athletic populations and potentially less healthy individuals, such as those with breathing illnesses, would also be of interest.

## 5. Conclusions

These data highlight an increased reliance on more efficient abdominal and thoracic musculature, and less recruitment of subclavian musculature, in young endurance athletes following a two-month breathing intervention. More efficient ventilatory muscular recruit at both lower and higher intensities during exercise may benefit endurance performance by reducing oxygen demand of the ventilator musculature and thus increasing oxygen availability for mechanical work.

## Figures and Tables

**Figure 1 jcm-10-03514-f001:**
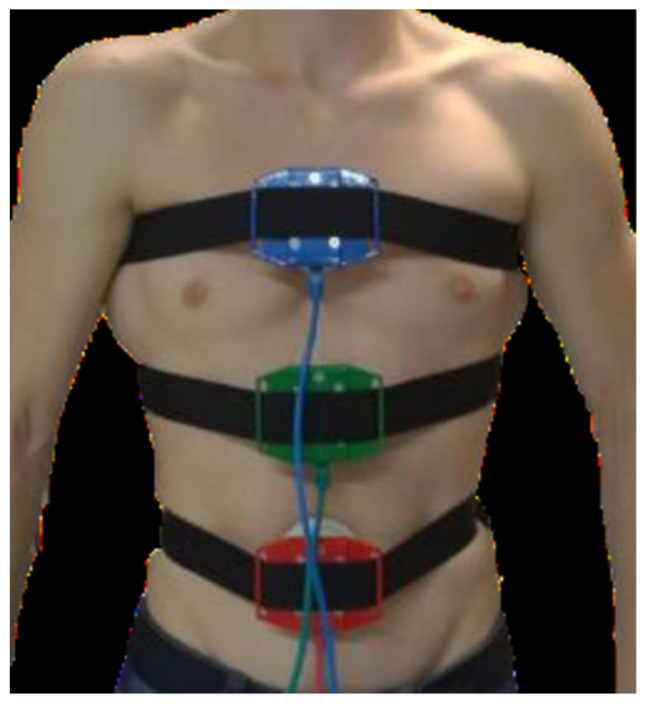
The positions of the location of probes on the body.

**Figure 2 jcm-10-03514-f002:**
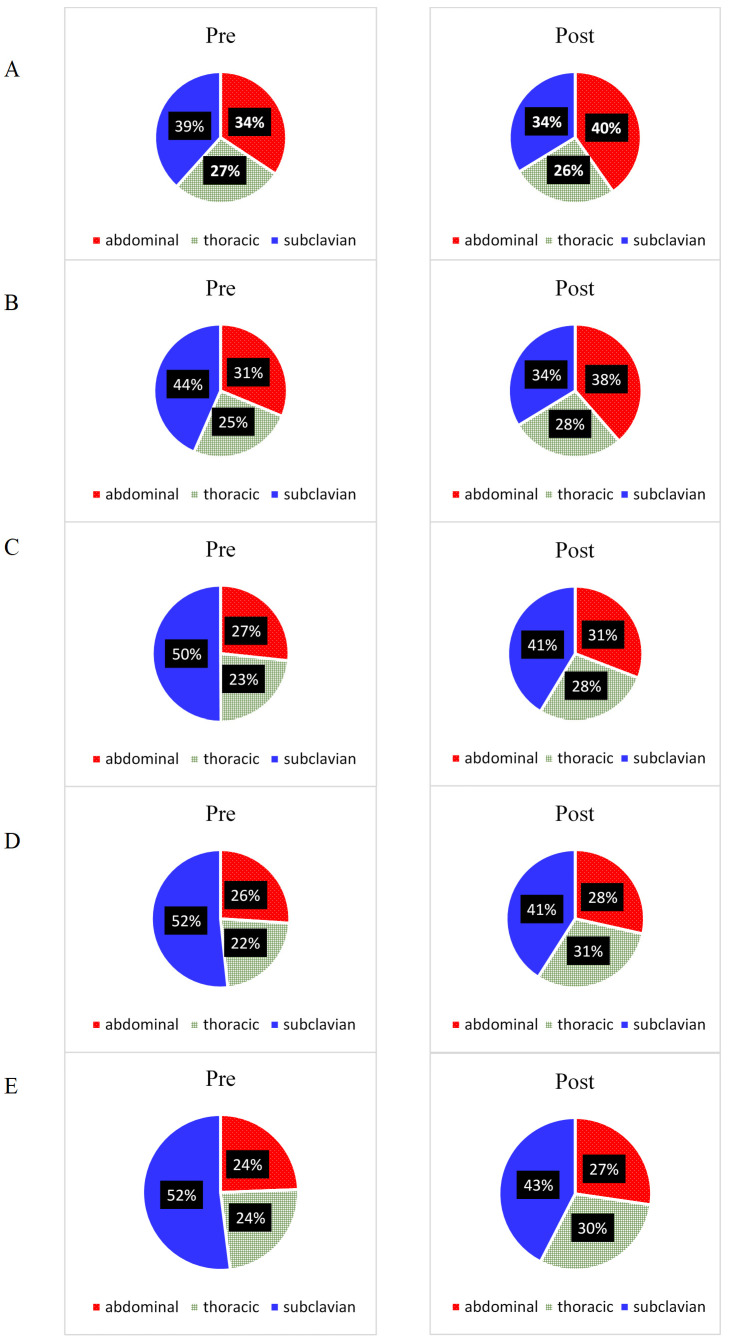
Engagement of breathing sectors at rest (**A**), during deep breathing (**B**), under load 2 W·kg^−1^ (**C**), under load 3 W·kg^−1^ (**D**), under load 4 W·kg^−1^ (**E**) pre and post intervention.

**Table 1 jcm-10-03514-t001:** Average values measured by probes and standard deviations of pressure on individual breathing sectors at rest and at different load intensities in the experimental group (EG) and control group (CG).

Breathing	Time	Rest	Deep–Rest	2 W·kg^−1^	3 W·kg^−1^	4 W·kg^−1^
Sector	[*n*·cm^−2^]	[*n*·cm^−2^]	[*n*·cm^−2^]	[*n*·cm^−2^]	[*n*·cm^−2^]	[*n*·cm^−2^]
abdominal	pre	0.54 ± 0.33	0.93 ± 0.36	1.38 ± 0.74	1.43 ± 0.66	1.54 ± 0.62
EG	post	0.94 ± 0.37 ^l^	1.79 ± 0.76 ^l,^**	2.01 ± 0.90 ^m,^*	2.04 ± 1.01 ^m,^*	1.91 ± 0.75 ^m^
chest	pre	0.46 ± 0.35	0.82 ± 0.56	1.32 ± 0.88	1.31 ± 0.75	1.59 ± 0.89
EG	post	0.65 ± 0.38 ^m^	1.34 ± 0.91 ^m^	1.79 ± 0.75 ^m^	2.19 ± 1.31 ^m,^**	2.16 ± 1.24 ^m^
subclavian	pre	0.65 ± 0.49	1.43 ± 0.79	2.92 ± 1.99	3.00 ± 1.39	3.59 ± 1.94
EG	post	0.83 ± 0.42 ^s^	1.66 ± 0.97 ^s^	2.70 ± 0.99	2.83 ± 0.88 *	3.08 ± 1.26 ^s^
abdominal	pre	0.56 ± 0.36	0.91 ± 0.40	1.33 ± 0.67	1.39 ± 0.55	1.55 ± 0.60
CG	post	0.57 ± 0.39	0.89 ± 0.42	1.35 ± 0.70	1.42 ± 0.52	1.54 ± 0.62
chest	pre	0.48 ± 0.34	0.85 ± 0.57	1.35 ± 0.90	1.36 ± 0.69	1.58 ± 0.90
CG	post	0.46 ± 0.37	0.88 ± 0.59	1.35 ± 0.85	1.37 ± 0.72	1.56 ± 0.88
subclavian	pre	0.60 ± 0.45	1.45 ± 0.82	2.95 ± 2.01	3.03 ± 1.43	3.49 ± 1.45
CG	post	0.63 ± 0.42	1.49 ± 0.88	2.91 ± 1.95	2.98 ± 1.35	3.40 ± 1.42

Note: * *p* < 0.05, ** *p* < 0.01, Cohen′s d: ^s^ small effect size, ^m^ medium effect size, ^l^ large effect size.

**Table 2 jcm-10-03514-t002:** Percent change in respiratory rate (RR), tidal volume (V_T_), minute ventilatory volume (V_E_) and oxygen consumption (VO_2_) after breathing exercises intervention versus exercise prior to breathing exercises intervention at different intensities in the experimental group (EG) and control group (CG).

		Rest	Deep-Rest	2 W·kg^−1^	3 W·kg^−1^	4 W·kg^−1^
EG	RR	−3.12 ^s^	−3.97 ^s^	−5.85 ^s^	−7.18 ^s,^*	−8.36 ^m,^**
	V_T_	-	-	10.60 ^s,^*	7.33 ^s^	6.00 ^s^
	V_E_	-	-	2.48	−0.60	−2.89
	VO_2_	-	-	−0.27	−0.15	−0.10
CG	RR	−1.32	−1.40	0.05	−0.07	0.45
	V_T_	-	-	−0.60	−0.15	0.07
	V_E_	-	-	1.72	−0.65	−0.31
	VO_2_	-	-	0.39	0.55	0.30

Note: ANOVA: * *p* < 0.05, ** *p* < 0.01, Cohen′s d: ^s^ small effect size, ^m^ medium effect size.

**Table 3 jcm-10-03514-t003:** Pearson correlation coefficient of change in tidal volume and abdominal probe pressure.

Correlation	2 W·kg^−1^	3 W·kg^−1^	4 W·kg^−1^
V_T_ and abdominal sector engagement	0.452 *	0.584 *	0.531 *

Note: * *p* < 0.05.

## Data Availability

Data sharing not applicable.

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
