# Peer review of "Impact of a Breathing Intervention on Engagement of Abdominal, Thoracic, and Subclavian Musculature during Exercise, a Randomized Trial"

_jcm, 2021, doi:10.3390/jcm10163514_

Round 1

Reviewer 1 Report

General comments: Dear Authors, The article “Impact of a breathing intervention on engagement of abdominal, thoracic, and subclavian musculature during exercise, a randomized trial ” is interesting, but several parts of the manuscript should be corrected. Specific comments: Introduction: The mail goal and hipothesis should be exhibit in separate paragraph. Materials and Methods: Study design is too long. Divide paragraphs on shorter and and sign titles for each. This approach is better for finding and understanding this section. Statistical analysis: Add information if you calculaled F coeefitient. Cohen's d or partial eta square for effect size determination in ANOVA? Results: Should be show in more visible.

Reviewer 2 Report

Abstract

  1. Line 22. Please change “lower” to “low”
  2. Since endurance exercise itself (i.e., higher minute ventilation) also involved in “breathing exercise”, I would like to suggest that change “breathing exercise” to any other appropriate words.

Introduction

  1. Line 34. Please reconsider delete of “many of”.
  2. Line41 and another Lines. Please change “total body” to “whole body” in the manuscript.
  3. Why did you chose breathing exercise based on yoga rather than inspiratory (respiratory) muscle training because respiratory muscle training is popular to improve respiratory function during endurance exercise.

Methods

  1. The subjects in the present study are endurance athletes. Did you control or check the actual their daily training volume (e.g., total running distance) during intervention period?
  2. Line 138. Why did you select this exercise protocol and exercise work load (W/kg)? You should add any appropriate reference.

Discussion

  1. Line232. Change “Inspiratory muscle” to “ inspiratory muscle”
  2. In the present study, authors presented the contribution of abdominal, thoracic, and subclavian muscle altered after the two month intervention of specific breath load training during endurance exercise. However, it is worth noting that to evaluate respiratory muscle fatigue and dyspnea during exercise as well as endurance performance when alter the contribution of abdominal, thoracic, and subclavian muscle after intervention period. Authors should mention about these points.

Author Response

Reviewer 2

Point 1: Abstract: Line 22. Please change “lower” to “low”.

Response 1: Thank you, corrected.

Point 2: Abstract: Since endurance exercise itself (i.e., higher minute ventilation) also involved in “breathing exercise”, I would like to suggest that change “breathing exercise” to any other appropriate words.

Response 2: Thank you. We edited as “We investigated whether a two-month yoga-based breathing intervention could affect breathing characteristics during exercise.” (lines 14-15).

Point 3: Introduction: Line 34. Please reconsider delete of “many of”.

Response 3: Deleted.

Point 4: Introduction: Line41 and another Lines. Please change “total body” to “whole body” in the manuscript.

Response 4: Corrected (Line 42, Line 50).

Point 5: Introduction: Why did you chose breathing exercise based on yoga rather than inspiratory (respiratory) muscle training because respiratory muscle training is popular to improve respiratory function during endurance exercise.

Response 5: This is a valid question. Previous research from our group has demonstrated an influence of a yoga-based breathing intervention on ventilation parameters during exercise. Building on this work, we aimed to determine whether this yoga-based breathing intervention would also influence breathing sector involvement. In future research it would be interesting to compare the effects of our yoga-based breathing intervention to traditional inspiratory muscle training.

Point 6: Methods: The subjects in the present study are endurance athletes. Did you control or check the actual their daily training volume (e.g., total running distance) during intervention period?

Response 6: Thank you for this point. Yes, we met every day with all of participants and their coach. We checked their daily training volume. All participants are members of the same training group, and thus training was similar for all athletes (Line: 99, 100).

Point 7: Methods: Line 138. Why did you select this exercise protocol and exercise work load (W/kg)? You should add any appropriate reference.

Response 7: Thank you for the question. This protocol was chosen so as to ensure the same relative load for all participants. Our group has successfully implemented this protocol in other investigations (10.7752/jpes.2020.06455 and 10.23736/s0022-4707.19.09483-0) (Line: 153). And next authors use similar work load (W/kg) (10.1123/ijspp.2019-0558)

Point 8: Discussion: Line232. Change “Inspiratory muscle” to “ inspiratory muscle”

Response 8: Corrected.

Point 9: Discussion: In the present study, authors presented the contribution of abdominal, thoracic, and subclavian muscle altered after the two month intervention of specific breath load training during endurance exercise. However, it is worth noting that to evaluate respiratory muscle fatigue and dyspnea during exercise as well as endurance performance when alter the contribution of abdominal, thoracic, and subclavian muscle after intervention period. Authors should mention about these points.

Response 9: Thank you for this comment. Indeed, our intervention was not tailored to evaluate the possible effects of such a breathing intervention during respiratory muscle fatigue or dyspnea, which may be relevant in clinical populations. We have added this to the discussion (lines 317-318)

English was checked and corrected by a native speaker.
